# Three-dimensional tracking of the ciliate *Tetrahymena* reveals the mechanism of ciliary stroke-driven helical swimming

Akisato Marumo[1], Masahiko Yamagishi [1] & Junichiro Yajima [1,2,3✉]

Helical swimming in free-space is a common behavior among microorganisms, such as ciliates that are covered with thousands hair-like motile cilia, and is thought to be essential for cells to orient directly to an external stimulus. However, a direct quantification of their three-dimensional (3D) helical trajectories has not been reported, in part due to difficulty in tracking 3D swimming behavior of ciliates, especially *Tetrahymena* with a small, transparent cell body. Here, we conducted 3D tracking of fluorescent microbeads within a cell to directly visualize the helical swimming exhibited by *Tetrahymena*. Our technique showed that *Tetrahymena* swims along a right-handed helical path with right-handed rolling of its cell body. Using the *Tetrahymena* cell permeabilized with detergent treatment, we also observed that influx of $Ca^{2+}$ into cilia changed the 3D-trajectory patterns of *Tetrahymena* swimming, indicating that the beating pattern of cilia is the determining factor in its swimming behavior.

[1] Department of Life Sciences, Graduate School of Arts and Sciences, The University of Tokyo, 3-8-1 Komaba, Meguro-ku, Tokyo 153-8902, Japan. [2] Komaba Institute for Science, The University of Tokyo, 3-8-1, Komaba, Meguro-ku, Tokyo 153-8902, Japan. [3] Research Center for Complex Systems Biology, The University of Tokyo, 3-8-1, Komaba, Meguro-ku, Tokyo 153-8902, Japan. ✉email: yajima@bio.c.u-tokyo.ac.jp

Ciliates are ciliated protists that swim in the water along helical paths by beating the thousands of cilia coating the cell body[1,2]. Helical swimming, which is typical for a swimming cell at low Reynolds number[2–5], is associated with its efficient movement for obtaining nutrients or for escaping from threats[6–10]. One such ciliate, *Tetrahymena*, has long been used as a model organism for a broad range of biological studies[11]. Its swimming pattern has been described to involve helical motions, as observed through classical microscopy, which is capable of imaging only one focal plane. However, *Tetrahymena* is a 3D entity, whose helical swimming trajectory is not limited to only one focal plane. In fact, the handedness of helical swimming trajectories of *Tetrahymena* has not been determined quantitatively, owing to the difficulty in measuring fast 3D motions (swimming velocity of approximately $500\,\mu\text{m s}^{-1}$)[12] of small (approximately $50\,\mu\text{m}$ in length) and transparent cell bodies[13]. The trajectory of the swimming cell appears to different observers to be following either a right- or left-handed helical path due to an optical illusion, sometimes referred to as the "spinning dancer[14,15]" caused by a lack of spatial visual cues[16]. Another ciliate, *Paramecium*, which is approximately 5 times larger than *Tetrahymena*, has been reported to swim along left-handed helical paths[1,17], and only two species of *Paramecium*, *P. calkinsi*, and *P. duboscqui*, have been reported to be right-handed helical swimmers, as observed under a classical microscope[18,19]. A precise determination of the 3D parameters of helical swimming motions of ciliates has not been conducted till date. 3D quantification of their trajectories would be very helpful for probing the swimming mechanism of ciliates.

Here we observed the 3D swimming behavior of ciliates *Tetrahymena* in free-space by 3D tracking of fluorescent microbeads phagocytosed by the cells. The 3D tracking was achieved using a three-dimensional prismatic optical tracking (termed *tPOT*) microscope[20], which yields 3D positional information. We report that ciliate *Tetrahymena thermophila* swims along right-handed helical paths, while the cell body rotates in a right-handed manner along its longitudinal axis. Furthermore, *Tetrahymena* cells permeabilized with detergent treatment, which swim backward with $Ca^{2+}$ stimulus[21,22], showed various helical swimming patterns, including backward swimming along right-handed helical paths, forward swimming along left-handed helical paths, or frequent changes in these two behaviors in response to $Ca^{2+}$ stimulus. Based on our study outcomes and available scientific literature, here, we discuss the possible mechanism underlying the formation of various helical swimming patterns formed by *Tetrahymena* in free-space.

## Results and discussion
**Direct observation of three-dimensional swimming trajectories of ciliate *Tetrahymena* in free-space.** To determine the handedness of helical swimming of ciliate *Tetrahymena* in free-space (Supplementary Movie 1), 3D swimming trajectories of *Tetrahymena* cells were tracked using a *tPOT* microscope, originally developed by us for studying molecular motor proteins using quantum dots[20], single fluorescent molecules[23], and microbeads[24]. We applied *tPOT* microscopy to a ciliate swimming assay by tracking fluorescent microbeads (200 nm in diameter) inside the cells after their intake by phagocytosis (Fig. 1a–c and Supplementary Movie 2). Using this method with high-spatial-resolution (Supplementary Fig. 1), we found *Tetrahymena thermophila* to have right-handed helical swimming motions (Fig. 1d). Trajectories on x–y and x–z planes indicated sinusoidal oscillations, the phase of oscillation on the x–y plane being faster by approximately 90° than that on the x–z plane (Fig. 1e), confirming right-handed helical motion (Fig. 1f). *T. thermophila* always swims along

right-handed helical paths ($n = 60$), whereas some cells swim straight or move randomly only after hitting the surface of a cover glass. An average helical pitch was estimated to be $129 \pm 29\,\mu\text{m}$ (mean ± standard deviation, SD) (Supplementary Fig. 2), which is robust to changes in the forward velocity (Supplementary Fig. 3). We also examined handedness of helical motions driven by *Tetrahymena pyriformis* ($n = 100$). Although *T. thermophila* and *T. pyriformis* are close evolutionary relatives among ciliates, they are fairly distant within species comprising the *T. pyriformis* complex[25]. Using the *tPOT* system, we found the helical motion with an average helical pitch of $132 \pm 43\,\mu\text{m}$ (mean ± SD) driven by *T. pyriformis* to show the same handedness as that observed for *T. thermophila* (Supplementary Fig. 4). Although forward and revolving velocities of *T. pyriformis* were higher than those of *T. thermophila*, helical pitches of the two were not significantly different ($p = 0.79$, Wilcoxson rank-sum test) (Supplementary Figs. 5 and 6), indicating that a right-handed helical swimming trajectory is surprisingly robust amongst related species. Moreover, we found other ciliates, *Paramecium multimicronucleatum* and *Paramecium calkinsi*, to show left- and right-handed helical trajectories (Supplementary Fig. 7), respectively, consistent with the previous report based on the observation under a classical optical microscope[1,17,18] (Supplementary Movie 3).

We next explored whether the cell rotates along its longitudinal axis (i.e., roll) during right-handed helical swimming, and if so, which direction it rotates in (i.e., rolling direction). Since the 3D trajectory of one point in the cell body represented the sum of helical swimming motion and cell body rolling, our method was applied to track two separate light spots in single *T. thermophila* cell, corresponding to fluorescent beads in separate food vacuoles, which is able to extract the cell body rolling via the relative position of two spots (Supplementary Fig. 8). We found that the two spots in a single cell showed right-handed helices with a constant phase difference; the position of the front spot with respect to the rear spot showed right-handed rotation over time, indicating that the cell body rolled in a right-handed manner while swimming forward (Fig. 1g, h). Thus, the rolling direction coincided with the handedness of the helical swimming trajectories as predicted in previous theoretical reports[2,3,26] and observed in other protists; *Paramecium*[27,28], *Chlamydomonas reinhardtii*[9,29] and *Euglena gracilis*[30].

**Three-dimensional observation of various swimming behaviors exhibited by *Tetrahymena* cells in response to depolarizing stimulation with $Ca^{2+}$.** In order to investigate the mechanism underlying the unidirectional helical swimming driven by *Tetrahymena*, we examined the 3D trajectories of *Tetrahymena*-driven backward swimming induced by calcium stimulation. An increase in intraciliary $Ca^{2+}$ in *T. thermophila* permeabilized with detergent treatment has been known to trigger backward swimming along the longitudinal cell axis[21,22] (hereafter, permeabilized cells with excess $Ca^{2+}$ are referred to as "doping cells"). On tracking the 3D trajectories of doping cells in the presence of $100\,\mu\text{M CaCl}_2$, we found that the backward swimming trajectories formed right-handed helices (Fig. 2a, blue), the handedness being same as that observed in forward swimming of non-doping cells (Supplementary Fig. 9 and Supplementary Movie 4). After adapting to the $Ca^{2+}$ stimulus (within approximately 5–30 s), the cells stopped helical swimming, along with rotation on spot for a few seconds (Fig. 2a, light-blue), and regained forward right-handed helical swimming (Fig. 2a, green to red), as noted previously[11]. In addition to the backward right-handed helical swimming after the stimulation, we observed the slow forward left-handed helical swimming of doping cells (Fig. 2b, blue to light-blue) before adaptation. Some doping cells switched back

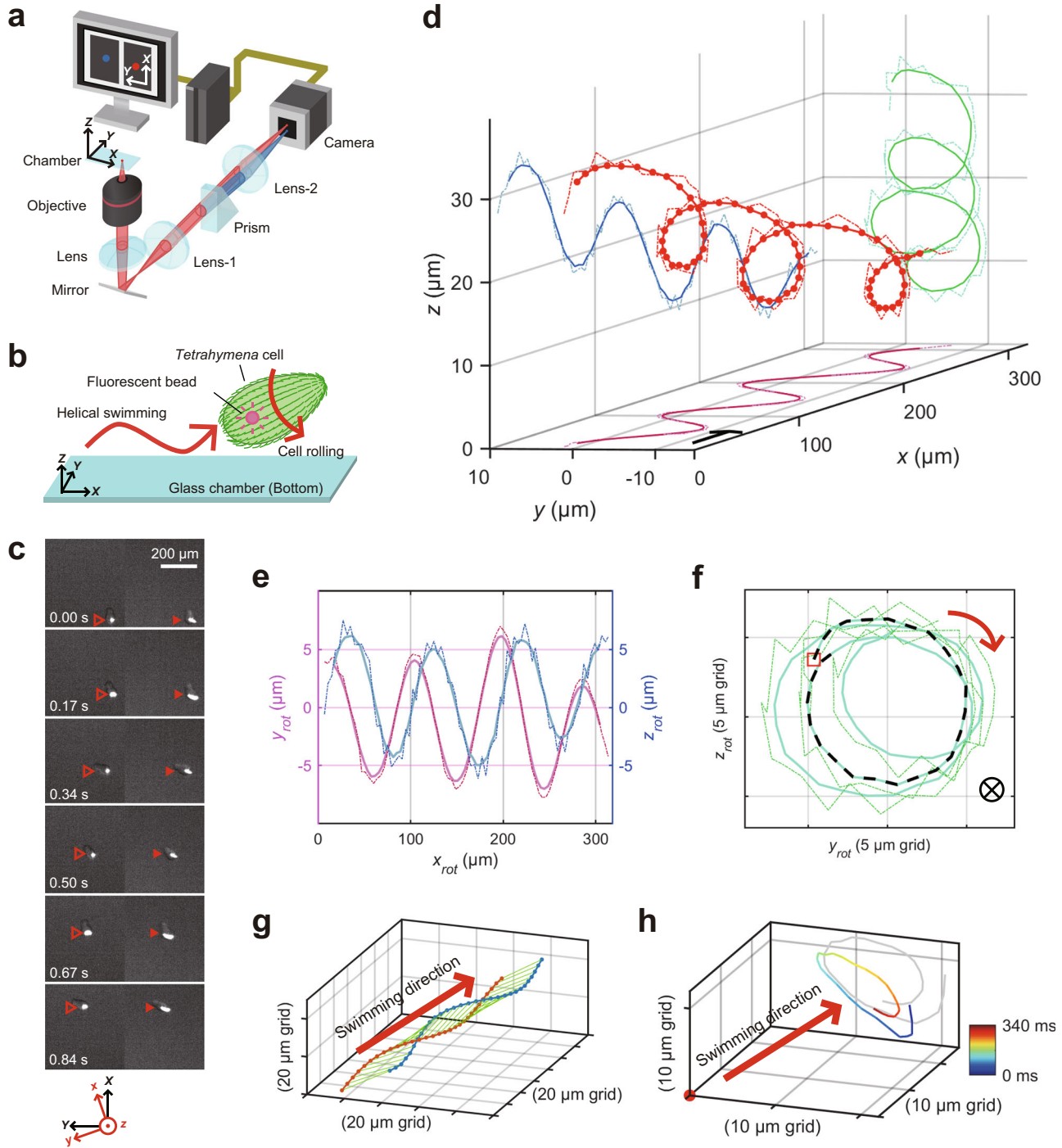

**Fig. 1 Three-dimensional swimming trajectory of T. thermophila. a** Diagram of a *tPOT* microscope (not to scale). **b** Diagram of *Tetrahymena* swimming after ingestion of a fluorescent bead (not to scale). **c** Sequential images of the bead in a swimming cell, observed using a *tPOT* microscope. The filled and open arrowheads indicate the images (split by the prism) of the bead in a moving cell, respectively (time in seconds). **d** 3D plot of the bead (red) in a cell revealed right-handed helical swimming of *T. thermophila*. Dotted lines show the data acquired at 89 frames s$^{-1}$, whereas the solid lines show the data averaged over every six frames. The arrow indicates the direction and distance of swimming during 100 ms along the *x*-axis. The *x–y* (pink) and *x–z* (blue) trajectories (**e**) and the *y–z* trajectory (**f**) of the bead in the cell in (**d**). Axes are rotated so that *x*-axis is parallel to the swimming direction. The trajectory of the first revolution is shown by the dotted black line and begins at the red open square (**f**). Based on this analysis, the handedness of helical swimming (red arrow) was checked. Forward velocity, revolving velocity, and helical pitch were 350 μm s$^{-1}$, 24 rad s$^{-1}$, and 92 μm, respectively. **g** Trajectories of two separate light spots in the same cell. The red and blue lines show the trajectories of the light spots, which were located posteriorly and anteriorly in the cell, respectively. **h** Relative trajectory of the anterior light spot when viewed from the posterior light spot (fixed at the origin).

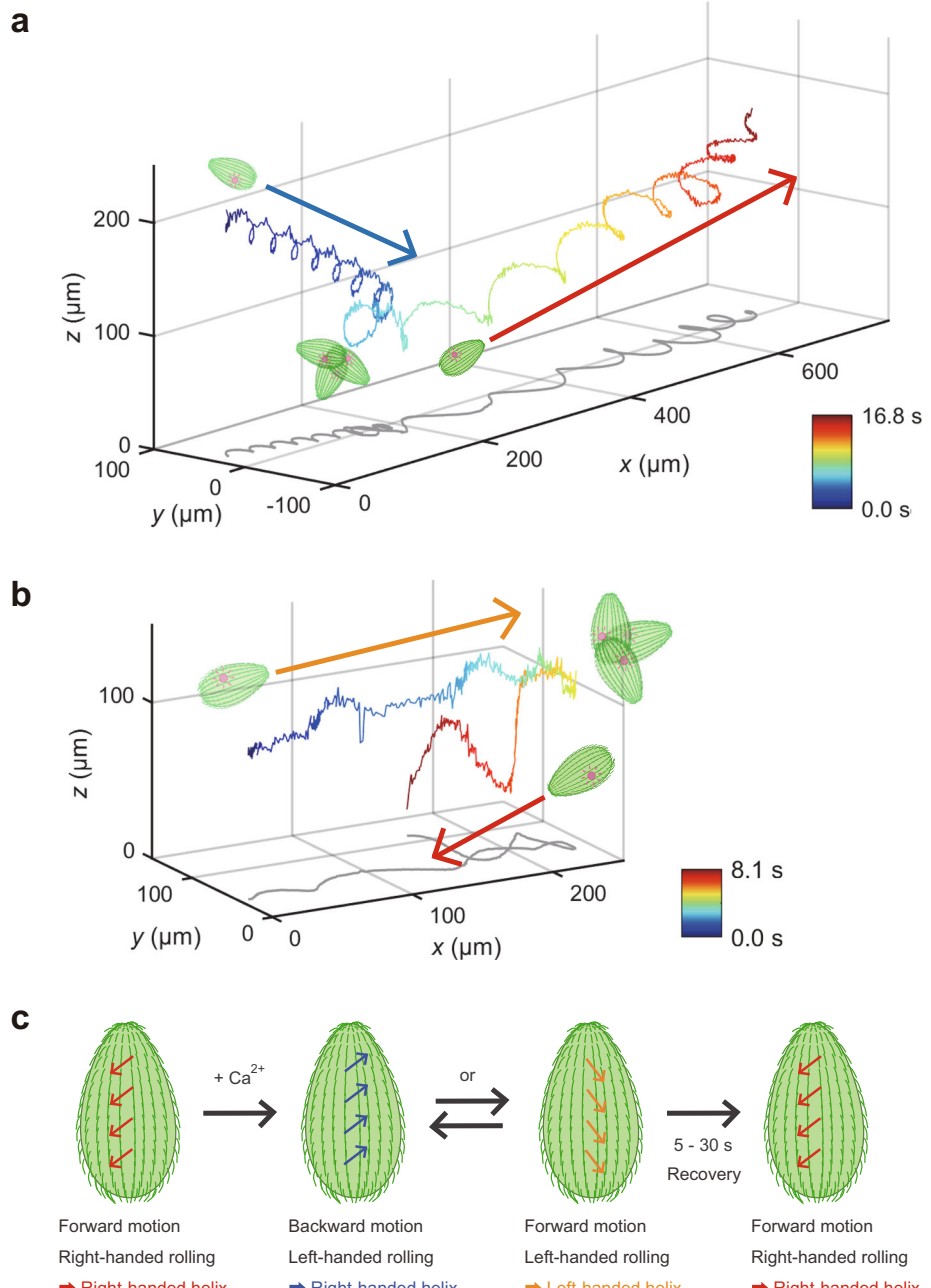

**Fig. 2 T. thermophila changed the swimming behavior when depolarizing stimulus was applied. a** An example of a 3D swimming trajectory of *T. thermophila* immediately after depolarizing stimulation with $Ca^{2+}$. The images represent the orientations of the cell body, and the blue and red arrows indicate the swimming directions. The colored line shows the data acquired at 89 frames $s^{-1}$. The cell swam backward along a right-handed helical path, and then rotated on the spot. After adaptation, the cell swam forward along a right-handed helical path. **b** Another example of 3D swimming trajectory with stimuli. The cell swam forward along a left-handed helical path for several seconds and rotated on the spot. Then, it swam forward along a right-handed helical path. **c** Bidirectional swimming model. The relationship between the effective stroke direction of cilia and the handedness of helical swimming of the cell is overviewed. Normally, *Tetrahymena* swam forward and along a right-handed path. However, when the effective stroke direction changed, their swimming behavior also altered, usually recovering within 30 s.

and forth between the two different motions (Supplementary Fig. 9 and Supplementary Movie 5) before adaptation. Thus, the patterns of helical swimming motion, driven by individual cells, changed in response to $Ca^{2+}$ stimulation.

As explained by Purcell, helical swimming trajectories are common among microorganisms moving at low Reynolds number[4]. For ciliate/flagellate protists, the motion of the cell at low Reynolds number could be driven by periodical whip-like beatings of cilia or flagella protruding from the cell body. The beat cycle consists of an effective stroke and a recovery stroke, which show distinct shapes[31] and force profiles[32]. Each beat causes a discrete translation followed by a rotation of the cell body in free-space, and the repeated translation and rotation in three-dimension induces a helical swimming trajectory of the cell[6–9,30], which has been generalized as the Helix Theorem[30,33,34]. Our direct 3D observation of the swimming behavior of the ciliate *Tetrahymena* showed that cells swam along right-handed helical paths with right-handed rolling (Fig. 1). This is probably due to the effective stroke direction of the

cilia, taking place from the right anterior direction to the left posterior direction[1]. The repeated effective stroke simultaneously produces the forward movement and rotation of the cell, as the forward swimming velocity and revolving velocity are tightly coupled (Supplementary Fig. 3a). Although our current system failed to detect the effect of periodic ciliary beating (Supplementary Fig. 10), a recent experiment with flagellate *Euglena gracilis* has observed the effect of periodic flagellar beating[30]. Using a method to observe the changes in the 3D shapes of cilia at a high spatiotemporal resolution, the effect of periodic ciliary beating on helical swimming motion would be detected.

Our 3D tracking technique also revealed that the swimming behavior could be regulated by intraciliary $Ca^{2+}$ perturbation (Fig. 2a, b). Since the mechanism of forward right-handed helical swimming remains unclear in case of the ciliate *Tetrahymena*, the mechanism by which depolarizing stimuli modulate chirality of the direction and helicity of swimming is also unclear. One possible explanation is as follows: the route of effective stroke can change, which determines not only forward/backward motion but also rolling direction (Fig. 2c)[1,26]. The repeated ciliary effective strokes towards the organism's rear left would cause right-handed helical forward swimming, as mentioned above. For backward swimming, the strokes would be opposite, as the handedness of the helix is the same as in forward swimming (Fig. 2a). In fact, a reversal of the beating direction of cilia has been observed to revert the swimming direction of *Paramecium*[35]. In the present study, after stimulation by $Ca^{2+}$, forward left-handed helical swimming (Fig. 2b) and frequent changes between right-handed backward and left-handed forward swimming (Supplementary Fig. 9 and Supplementary Movie 5) were also observed unexpectedly. This suggested that the handedness of helical swimming is not decided by the cell shape alone. Instead, a changeable route of the effective stroke is the key determinant of swimming behavior. The 3D orbits of ciliary strokes can change after $Ca^{2+}$ stimulation[21,36], and left-handed forward swimming is interpreted as an intermediate state of right-handed forward and backward swimming. Models addressing the mechanism by which an individual cilium produces directed force will be tested in the near future with further 3D studies combined with computational analysis.

## Methods

**Cell culture**. Strain SB 255 of *T. thermophila* and strain W of *T. pyriformis* were cultured in 50 ml of SPP medium [1% Proteose Peptone No. 3, 0.2% glucose, 0.1% yeast extract, 0.003% Fe-EDTA, and 1% antibiotic antimycotic solution] (and 0.5% paromomycin, only for *T. thermophila*) in 125 ml bottles in an incubator at 15 °C. Serial transfers of the cells were performed once in every two weeks. The cells were transferred to another incubator at 30 °C the day before the observation.

**Experimental setting**. Before observation, 500 μl of culture medium containing cells was centrifuged at 400 × g for 1 min at ~25 °C, and 450 μl supernatant was discarded. Then, 5 μl of 0.2 μm fluorescent beads solution, 0.01% solids (F8810, Thermo Fisher Scientific) was added. The fluorescent beads are thought to be orally phagocytosed into food vacuoles at the base of the oral apparatus and further transported into the cell[37]. After 5 min, 950 μl of M-buffer [10 mM tricine, 0.5 mM MOPS, 50 μM CaCl₂, 8 mM NaCl, pH 7.4] were added and the mixture was centrifuged at 400 × g for 1 min at ~25 °C. Next, 850 μl of the supernatant was discarded, and the cells were resuspended in 850 μl of new M-buffer. The suspension was centrifuged at 400 × g for 1 min at ~25 °C, and the steps were repeated five times. Finally, 950 μl of supernatant was discarded and after a five min incubation, the mixture containing cells was loaded into an observation chamber. The observation chamber consisted of two cover glasses (NEO Micro cover glass; Thickness No. 1; bottom: 24 × 36 mm, top: 18 × 18 mm, Matsunami, Tokyo, Japan) and four layers of double-sided tape. The height of the chamber was approximately 400 μm, which was much larger than the size of *Tetrahymena* cells. To prevent cell adsorption, the chamber was filled with 20 μl of 5 mg ml⁻¹ Bovine serum albumin (Sigma-Aldrich) for 5 min and finally washed with 100 μl of M-buffer before use. Assays were carried out at 22.5 ± 1 °C.

**Doping cell**. The 0.2 μm fluorescent beads were ingested by the cells, as described in the "Experimental setting" section. For the preparation and observation of doping cells, KCl-buffer [50 mM KCl, 10 mM Tris-maleate (pH 7.0), 0.5% polyethylene glycol] was used instead of M-buffer. To make a hole in the cell membrane and allow $Ca^{2+}$ to flow into the cells, 50 μl of M-Triton-buffer [10 mM tricine, 0.5 mM MOPS, 100 μM CaCl₂, 8 mM NaCl, 0.0167% Triton-X, pH 7.4] were poured onto 5 μl of the mixture containing cells on the cover glass (NEO Micro cover glass; Thickness No. 1; 24 × 36 mm, Matsunami, Tokyo, Japan). Observations were performed within 30 s after the addition of M-Triton-buffer. In this study, the swimming trajectories of 16 doping cells were obtained; three cells swam backward along right-handed helical paths, six cells swam forward along left-handed helical paths, five cells swam switching continuously between the two swimming patterns, and two cells swam forward along right-handed helical paths.

**3D prismatic optical tracking microscope**. The swimming cells were observed under the 3D prismatic optical tracking (tPOT) microscope[20], which provides z-positional information from planar images with submicron accuracy. tPOT microscope uses a prism to split one image into two and calculates z-positional information of the sample from the difference in y-positions of the two images. As illustrated in Fig. 1a, the back-focal-plane (BFP) of the objective (UPLXAPO 10×, NA 0.4 or 4×, NA 0.16, Olympus, Tokyo, Japan) was focused outside the camera port of an inverted microscope (IX70, Olympus, Tokyo, Japan) with achromatic Lens-1 (combined focal length 170 mm) to make an equivalent BFP (eBFP). To split the image beam path at the eBFP, a custom-made wedge prism (94.8°, Natsume Kougaku, Nagano, Japan) coated with an antireflective layer was precisely located at the eBFP. The two split images of a sample were focused on the camera focal plane by achromatic Lens-2 (combined focal length 170 mm). Images were recorded by CMOS camera (C14440-20UP, Hamamatsu Photonics, Hamamatsu, Japan) via HSR software (Hamamatsu Photonics) on a Windows 10 PC at 10.00 or 11.21 ms per frame (2 × 2 binning). For calibration of z-axis position, a custom-built stable stage (Chuukousha Seisakujo, Tokyo, Japan) equipped with a pulse motor (SGSP-13ACTR-BO, Sigma Koki, Tokyo, Japan) and controller (QT-CM2, Chuo Precision Industrial, Tokyo, Japan) was used to move the objective vertically while observing the stable fluorescent beads (0.2 μm or 0.5 μm in diameter, F8812, Thermo Fisher Scientific) placed inside the observation chamber filled with 0.2% agarose gel (Agarose S, Nippon Gene, Tokyo, Japan) or attached to the bottom of the chamber filled with M-buffer. The calculated z-position and actual z-position (as defined by the pulse motor) corresponded linearly over a range of ±50 μm from the focal plane, and there was no significant difference in the results of calibration by these methods.

**Parameter calculation**. The forward and revolving velocities were calculated by fitting the x displacement−time plots (averaged over every six frames) and revolution−time plots (averaged over every six frames) with a linear function using MATLAB. Helical pitches were calculated as forward velocities divided by the revolving velocities. Individual data are shown in Supplementary Figs. 2 and 5.

**Statistics and reproducibility**. Statistical analysis of data was done using MATLAB. Wilcoxon rank-sum test were used to compare the forward velocities, revolving velocities, and helical pitches, respectively, of *T. thermophila* and *T. pyriformis* (Supplementary Fig. 6). Tracking data for beads within the swimming cell were obtained from at least three independent experiments, and sample sizes are indicated in detail in the text or the "Methods" section.

**Reporting summary**. Further information on research design is available in the Nature Research Reporting Summary linked to this article.

## Data availability
All samples used in this study are available from the corresponding authors on request. The source data for graphs in the main figures are provided as Supplementary Data 1.

## Code availability
Custom scripts were written for tracking beads within swimming cells with Igor Pro 5.05A, and for analysis of trajectories with MATLAB R2021a. These scripts are available on reasonable request.

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

## Acknowledgements

We thank Kyohei Matsuda, Mitsuhiro Sugawa, and Yoko Y. Toyoshima for critical discussion. *T. pyriformis* was kindly gifted by Kentaro Nakano, University of Tsukuba, Japan. *Paramecium* strains were provided by NBRP Paramecium lab, Yamaguchi Univ. with support in part by MEXT National Bio-Resource Project. This work was supported in part by JSPS KAKENHI (grant numbers JP20K06635, JP21K19252 to J.Y.); MEXT KAKENHI Grant-in-Aid for Scientific Research on Innovative Areas (grant numbers JP19H05357, JP21H00386 to J.Y.).

## Author contributions

J.Y. designed the project. A.M. and M.Y. prepared biological samples. A.M. performed and analyzed behavioral bioassays. All authors wrote the manuscript.

## Competing interests

The authors declare no competing interests.
