## [Transparent Peer Review File · Communications Biology]

Reviewers' comments:

Reviewer #1 (Remarks to the Author):

Report on the paper:

Three-dimensional tracking of the ciliate *Tetrahymena* to probe the mechanism of ciliary stroke-driven helical swimming

by A. Marumo, M. Yamagishi, and J. Yajima

The paper contains interesting new quantitative observations of the swimming trajectories of the ciliate *Tetrahymena*.

The content of the paper is interesting, the conclusions are well supported by the data, and the paper is well written.

There are however two main concerns that, in my opinion, the authors should address before publication can be recommended.

1. I believe that the readers of the manuscript should be made aware that recent biophysical literature argues that helical trajectories of ciliates are, in fact, simply the hydrodynamic signature of the periodic beating of their array of cilia (or of their flagella in the case of flagellates).

This is a consequence of translational and rotational symmetry of hydrodynamics of low Reynolds numbers swimmers that move in a homogenous medium and far away from the influence of external walls.

In other words, hydrodynamics at low Reynolds number dictates that *every* microscopic swimmer that moves thanks to periodic shape changes traces a helical trajectory (in a sense that is made precise in Ref. [1] below).

This very general fact is known as "The Helix Theorem" and discussed at length in Ref. [1]. It is stated and discussed also in references [2] and [3] (possibly in a slightly less obvious form due to use of a more technical language). This statement does not diminish at all the value of the findings by the authors, which consist of new experimental observations of a biological organism that could be used to either prove or disprove the general theoretical result in a specific situation.

2. The authors reconstruct helical trajectories on the basis of time-discrete snapshots, which are then averaged.

For example, in Figure 1d, data are acquired at 89 frames per second, and then averaged over every 6 frames.

Data show fine oscillations which are then smoothed out by averaging.

I wonder about the meaning of the finer oscillations of the trajectories. Are they associated with noise, or do they encode instead a faster dynamics with a time period much smaller than the time it takes to the organism to complete one full helical turn?

How do these slow and fast time scales correlate with the time scale of the ciliary beat?

I wonder whether these faster time scales are sufficiently well resolved or whether observations at a faster sampling rate would reveal meaningful structure and finer features in the helical trajectory, which is sampled only at discrete time steps.

And I wonder whether the authors are recognizing the potential value of these fine scale features.

They are not discussed in the manuscript, which instead focuses only on the smooth helices resulting from the averaging of the trajectories.

Ref. 2 below could provide some guidance in interpreting these fine features.

Some additional minor points are listed below.

-line 46

have been reported to be right-handed helical swimmers?

-legend of Fig.2

What is the meaning of right/left rotation? Wouldn't right-handed/left-handed rotation be clearer?

-line 95

"will" instead of "would"?

-line 116 and following: unclear

What is the meaning of "torque generating strokes backwards and to the organism's left?"

What is the meaning of "oblique orientation of torque generating stroke"?

What is the "orientation of a stroke"?

-line 188

Statistics instead of Statics?

Additional references

1. G. Cicconofri and A. DeSimone: Modeling biological and bio-inspired swimming at microscopic scales: recent results and perspectives. *Computers & Fluids*, 179, 799-805 (2019).
2. M. Rossi et al.: Kinematics of flagellar swimming in *Euglena gracilis*: helical trajectories and flagellar shapes. *Proceedings of the National Academy of Sciences USA* 114(50), 13085-13090 (2017).
3. A. Shapere and F. Wilczek: Geometry of self-propulsion at low Reynolds number. *Journal of Fluid Mechanics* 198, 557-585 (1989).

Reviewer #2 (Remarks to the Author):

This manuscript reports experimental measurements of 3d motion of ciliate *Tetrahymena*. The authors use a custom-made microscope to track the 3d trajectories of fluorescent beads ingested by the cells and reconstruct the detailed 3d motion of the cells from beads data. They showed that *Tetrahymena* swims along a right-handed helical path with right-handed rolling; influx of Ca^{2+} into cilia caused different patterns of 3D-trajectories of *Tetrahymena* swimming. I think the authors reported interesting data in the manuscript and recommend its publication after the following comments are addressed.

1. The authors should say more about the process through which the cells ingest fluorescent beads. The supporting videos seem to show that, in many cases, beads appear in the head region of the cell. Is there a reason for that?
2. There are previous studies that use conventional microscopes to reconstruct 3d motion of algae, for example: <https://www.pnas.org/content/114/50/13085.abstract> . The authors should cite these papers.

Our response to the reviewers' comments

Original Reviewers' comments

Our response

Reviewer #1 (Remarks to the Author):

Report on the paper:

Three-dimensional tracking of the ciliate Tetrahymena to probe the mechanism of ciliary stroke-driven helical swimming by A. Marumo, M. Yamagishi, and J. Yajima

The paper contains interesting new quantitative observations of the swimming trajectories of the ciliate Tetrahymena.

The content of the paper is interesting, the conclusions are well supported by the data, and the paper is well written.

There are however two main concerns that, in my opinion, the authors should address before publication can be recommended.

*1. I believe that the readers of the manuscript should be made aware that recent biophysical literature argues that helical trajectories of ciliates are, in fact, simply the hydrodynamic signature of the periodic beating of their array of cilia (or of their flagella in the case of flagellates). This is a consequence of translational and rotational symmetry of hydrodynamics of low Reynolds numbers swimmers that move in a homogenous medium and far away from the influence of external walls. In other words, hydrodynamics at low Reynolds number dictates that *every* microscopic swimmer that moves thanks to periodic shape changes traces a helical trajectory (in a sense that is made precise in Ref. [1] below). This very general fact is known as "The Helix Theorem" and discussed at length in Ref. [1]. It is stated and discussed also in references [2] and [3] (possibly in a slightly less obvious form due to use of a more technical language). This statement does not diminish at all the value of the findings by the authors, which consist of new experimental observations of a biological organism that could be used to either prove or disprove the general theoretical result in a specific situation.*

This is an exceptionally insightful comment, and we have now re-organised the discussion around the reviewer's suggestion on explaining the data by simply using the hydrodynamic signature of the periodic beats of the cilia. The discussion is now an exploration of the extent to which this periodic beat (consisting of an effective stroke and a recovery stroke) can explain the data. We think that this is a major improvement, and we would like to thank the reviewer for their insight.

(pages 7-8, lines 98-109)

As explained by Purcell, helical swimming trajectories are common among microorganisms moving at a low Reynolds number²¹. For ciliate/flagellate protists, the motion of the cell at a low Reynolds number could be driven by a periodical whip-like beating of cilia or flagella protruding from the cell body. The beat cycle consists of an effective stroke and a recovery stroke that show distinct shapes²² and force profiles²³. Each beat causes a discrete translation followed by a rotation of the cell body in free-space, and the repeated translation and rotation in three-dimension induces a helical swimming trajectory of the cell^{4,18}, which has been generalized as the Helix Theorem^{18,24,25}. Our direct 3D observation of the swimming behavior of the ciliate *Tetrahymena* showed that cells swam along right-handed helical paths with right-handed rolling (Fig. 1). This is probably due to the effective stroke direction of the cilia, taking place from a right anterior direction to the left posterior direction¹. The repeated, effective stroke simultaneously produces the forward movement and rotation of the cell, as the forward swimming velocity and revolving velocity are tightly coupled (Supplementary Fig. 3a).

(page 9, lines 120-123)

One possible explanation is as follows: the route of effective stroke can change, which determines not only forward/backward motion but also rolling direction (Fig. 2c)^{1,16}. The repeated ciliary effective strokes towards the organism's rear left would cause right-handed helical forward swimming, as mentioned above.

2. The authors reconstruct helical trajectories on the basis of time-discrete snapshots, which are then averaged. For example, in Figure 1d, data are acquired at 89 frames per second, and then averaged over every 6 frames. Data show fine oscillations which are then smoothed out by averaging. I wonder about the meaning of the finer oscillations of the trajectories. Are they associated with noise, or do they encode instead a faster dynamics with a time period much smaller than the time it takes to the organism to complete one full helical turn?

How do these slow and fast time scales correlate with the time scale of the ciliary beat?

I wonder whether these faster time scales are sufficiently well resolved or whether observations at a faster sampling rate would reveal meaningful structure and finer features in the helical trajectory, which is sampled only at discrete time steps.

And I wonder whether the authors are recognizing the potential value of these fine scale features. They are not discussed in the manuscript, which instead focuses only on the smooth helices resulting from the averaging of the trajectories.

Ref. 2 below could provide some guidance in interpreting these fine features.

We highly appreciate the reviewers' insightful suggestion to analyze the raw data and to extract the finer oscillation that may correspond to the ciliary beat. However, our current imaging setup (*tPOT* + cMOS camera) is not capable of extracting the faster periodicity of ciliary beating from existing data. We have now inserted an extra supplementary figure (Supplementary Fig. 10) showing the raw data of *x-y*- and *x-z*-trajectories produced by *Tetrahymena thermophile*. Considering the limitation of our current experimental setup, it would make sense only if we could focus on the three-dimensional averaged trajectories of the cells and the determination of the handedness of its helical motion. Although we agree with this reviewer on the importance and potential value of measurement at a faster sampling rate as basic characteristics of swimming protists, this would need a lot of effort and is beyond the scope of our current work. We have now discussed this point and cited Ref. 2 (Rossi et al., *PNAS* 2017) as kindly notified.

(page 8, lines 110-114)

Although our current system failed to detect the effect of periodic ciliary beating (Supplementary Fig. 10), recent experiments with flagellate *Euglena gracilis* have observed the effect of periodic flagellar beating¹⁸. Using a method to observe the changes in the three-dimensional shapes of cilia at a high spatiotemporal resolution, the effect of periodic ciliary beating on helical swimming motion would be detected.

(Supplementary Fig. 10)

Raw data of *x-y* (pink) and *x-z* (blue) trajectories of the beads during *T. thermophila* cell swimming in free-space. The trajectories of 8 cells are shown. Data were acquired at 89 frames/s. Axes are rotated so that the *x*-axis is parallel to the swimming direction. However, the effect of periodic ciliary beating was not detected by these trajectories.

Some additional minor points are listed below.

We thank the reviewer for pointing these out.

-line 46

have been reported to be right-handed helical swimmers?

“...to be a right-handed a helical swimmer...” was corrected to be “... to be right-handed helical swimmers...”

-legend of Fig.2

What is the meaning of right/left rotation? Wouldn't right-handed/left-handed rotation be clearer?

We removed original words and now mentioned the handedness of rolling motion of the cell.

-line 95

"will" instead of "would"?

"... would be referred to..." was corrected to be "... are referred to..."

-line 116 and following: unclear

What is the meaning of "torque generating strokes backwards and to the organism's left?"

What is the meaning of "oblique orientation of torque generating stroke"?

What is the "orientation of a stroke"?

We agree that our original words were unclear. We have made corrections.

-line 188

Statistics instead of Statics?

"Statics..." was corrected to be "Statistics..."

Additional references

- 1. G. Cicconofri and A. DeSimone: Modeling biological and bio-inspired swimming at microscopic scales: recent results and perspectives. Computers & Fluids, 179, 799-805 (2019).*
- 2. M. Rossi et al.: Kinematics of flagellar swimming in Euglena gracilis: helical trajectories and flagellar shapes. Proceedings of the National Academy of Sciences USA 114(50), 13085-13090 (2017).*
- 3. A. Shapere and F. Wilczek: Geometry of self-propulsion at low Reynolds number. Journal of Fluid Mechanics 198, 557-585 (1989).*

We appreciate that the reviewer kindly notified us that we failed to cite these three papers. These are now cited in the manuscript.

Reviewer #2 (Remarks to the Author):

This manuscript reports experimental measurements of 3d motion of ciliate Tetrahymena. The authors use a custom-made microscope to track the 3d trajectories of fluorescent beads

ingested by the cells and reconstruct the detailed 3d motion of the cells from beads data. They showed that Tetrahymena swims along a right-handed helical path with right-handed rolling; influx of Ca²⁺ into cilia caused different patterns of 3D-trajectories of Tetrahymena swimming. I think the authors reported interesting data in the manuscript and recommend its publication after the following comments are addressed.

We appreciate the reviewer for their high evaluation of our work.

1. The authors should say more about the process through which the cells ingest fluorescent beads. The supporting videos seem to show that, in many cases, beads appear in the head region of the cell. Is there a reason for that?

We agree that we failed to properly explain the process through which *Tetrahymena* ingests fluorescent beads. We now added the sentences as requested and cited a suitable reference.

“The fluorescent beads are thought to be orally phagocytosed into food vacuoles at the base of the oral apparatus and further transported into the cell (Guerrier et.al., *Traffic*, 2016)”

(page 10, lines 147-149)

2. There are previous studies that use conventional microscopes to reconstruct 3d motion of algae, for example: <https://www.pnas.org/content/114/50/13085.abstract> . The authors should cite these papers.

We appreciate that the reviewer notified us that we failed to cite Rossi *et al.* (2017). The reference is now cited in the manuscript.